# Genomic and Transcriptional Profiles of *Kelch-like* (*klhl*) Gene Family in Polyploid *Carassius* Complex

**DOI:** 10.3390/ijms24098367

**Published:** 2023-05-06

**Authors:** Fang Peng, Li Zhou, Weijia Lu, Ruihai Gan, Meng Lu, Zhi Li, Xiaojuan Zhang, Yang Wang, Jianfang Gui

**Affiliations:** 1State Key Laboratory of Freshwater Ecology and Biotechnology, Hubei Hongshan Laboratory, The Innovation Academy of Seed Design, Institute of Hydrobiology, Chinese Academy of Sciences, Wuhan 430072, China; 2University of Chinese Academy of Sciences, Beijing 100049, China

**Keywords:** *Kelch-like* gene, *Carassius*, polyploid, homeolog, allele

## Abstract

Genome duplication supplies raw genetic materials and has been thought to be essential for evolutionary innovation and ecological adaptation. Here, we select *Kelch-like* (*klhl*) genes to study the evolution of the duplicated genes in the polyploid *Carassius* complex, including amphidiploid *C. auratus* and amphitriploid *C. gibelio*. Phylogenetic, chromosomal location and read coverage analyses indicate that most of *Carassius klhl* genes exhibit a 2:1 relationship with zebrafish orthologs and confirm two rounds of polyploidy, an allotetraploidy followed by an autotriploidy, occurred during *Carassius* evolution. The lineage-specific expansion and biased retention/loss of *klhl* genes are also found in *Carassius*. Transcriptome analyses across eight adult tissues and seven embryogenesis stages reveal varied expression dominance and divergence between the two species. The expression of *klhls* in response to *Carassius* herpesvirus 2 infection shows different expression changes corresponding to distinct herpesvirus resistances in three *C. gibelio* gynogenetic clones. Finally, we find that most *C. gibelio klhl* genes possess three alleles except eight genes that have lost one or two alleles due to genome rearrangement. The allele expression bias is prosperous for *Cgklhl* genes and varies during embryogenesis owning to the sequential expression manner of the alleles. The current study provides global insights into the genomic and transcriptional evolution of duplicated genes in a given superfamily resulting from multiple rounds of polyploidization.

## 1. Introduction

The *Kelch-like* (*klhl*) gene family encodes a class of substrate adapter proteins for Cullin-E3 ubiquitination and is characterized by a broad-complex, tramtrack, bricá-brac/poxvirus and zinc finger (BTB/POZ) domain, a BACK domain, and five to six Kelch motifs. A total of 42 KLHL genes have been identified in the human genome [1]. They share similar secondary structures, but their primary sequences have little homology. Most KLHL proteins can interact with Cullin3 to form Cullin-E3 ubiquitin ligase complex via BTB/POZ domain and act as adapters to recruit substrates for ubiquitination through Kelch motifs. Although functions of many KLHL proteins remain unexplored, they have been considered crucial regulators in a variety of fundamental cellular processes [2,3,4]. In recent years, increasing evidence has shown that loss-of-function mutations and dysregulated expression of *KLHLs* are closely associated with cancers, hereditary diseases, and chronic disorders, resulting in *KLHLs* attracting considerable attention as potential therapeutic targets [4,5,6,7,8]. Up to now, research and information on the fish *klhl* gene family are scarce. Only a few members, such as *keap1* (*kelch-like ECH-associated protein 1*, also known as *klhl19*), *klhl15*, *klhl31*, and *klhl41* have been identified from several fish species, and the majority of these studies focused on the expression of *keap1* and its substrate nuclear *factor erythroid 2-related factor 2* in response to dietary nutrients or environmental contaminants to assess the antioxidant activities.

Besides the two rounds of whole genome duplication (WGD) that occurred at the root of vertebrates (1R and 2R), it has been proposed that a teleost-specific third WGD (Ts3R) occurs in the ancestor of teleost [9,10,11]. Additionally, polyploidization has independently and repeatedly occurred in several taxonomic fish orders [9,10,11]. For example, about 400 polyploid species across over 30 genera have been discovered in Cyprininae [12]. WGD has been considered a main driving force of speciation owing to its profound impacts on evolutionary innovation and ecological adaptation [13,14,15,16]. One consequence of WGD is expression alteration in entire networks involved in hundreds to thousands of duplicated genes, termed “expression evolution” or “regulatory evolution” [17]. Although the expression evolution after WGD has been well documented in paleopolyploids [13,18], we still have a limited understanding of young polyploids with complex genomes.

*Carassius* is an extraordinary genus, including multiple-ploidy organisms that reproduce by different reproduction modes [13,18]. Of them, sexual *C. auratus* (crucian carp/goldfish) is an amphidiploid (also called allotetraploid) (2*n* = 4*x* = 100, AABB), and gynogenetic *C. gibelio* (gibel carp) is an amphitriploid (previously called allo-/autohexaploid) (3*n* = 6*x* = ~150, AAABBB) that contains two subgenomes, and each subgenome has three homologs (Figure 1A) [19,20,21]. *Carassius* is distributed widely across the Eurasian continent [22,23,24] and is one of the most important aquaculture species in China [25,26,27]. In addition to Ts3R, it has been speculated that two extra rounds of polyploidy, an early allotetraploidy 10.17 million years ago (Mya) and a later autotriploidy 0.82 Mya, occurred during *Carassius* evolution (Figure 1A) [19]. After autotriploidy, *C. gibelio* evolved unique multiple reproduction modes and sex determination mechanisms [28,29]. Therefore, the *Carassius* complex provides a valuable chance to inspect the fates of duplicated genes and their impacts on evolutionary innovation and ecological adaptation. In this study, we explored the evolutionary trajectories of the *klhl* gene family in *Carassius* and investigated their expression panorama after experiencing multiple rounds of polyploidy.

## 2. Results

### 2.1. Genome-Wide Identification of klhl Genes

To explore the evolutionary origin of the *klhl* gene family, we retrieved *klhl* genes via a hierarchical strategy from the assemblies of *C. auratus* and *C. gibelio*, as well as from other 11 animals (Figure 1B). None of the *klhl* genes was identified from the genome of Cephalochordata amphioxus (*Branchiostoma belcheri*), a living invertebrate close to vertebrates [30]. Sea lamprey (*Petromyzon marinus*), a representative of the most ancient vertebrate [31,32], had the fewest *klhl* genes (20), suggesting that *klhl* genes would emerge during early vertebrate evolution.

Spotted gar (*Lepisosteus oculatus*), a teleost sister lineage that did not experience the Ts3R [33], had 44 *klhl* genes, whereas six teleosts analyzed in this study all possessed more *klhl* genes. For example, 67 *klhl* genes were identified in the zebrafish (*Danio rerio*) genome, of which 20 members had at least two duplicates that would be generated by either Ts3R or small-scale duplication. As paleo-tetraploids [34,35], rainbow trout (*Oncorhynchus mykiss*), and Atlantic salmon (*Salmo salar*) possessed 67 and 73 *klhl* genes in their genomes, respectively. Although the numbers were comparable to those of zebrafish, some *klhl* genes still had one more copy in these two *Salmonidae* fishes (Appendix A). In contrast, 96 and 98 *klhl* genes were identified in *C. auratus* and *C. gibelio*, respectively, and most of them had one more duplicate (Appendix A). It was confirmed that the two assemblies are both composed of one haplotype of two subgenomes (AB). Although possessing similar gene numbers, minor allele frequencies of single nucleotide polymorphism (SNP) sites in *klhl* genes were around 0.34 (approximately 1/3) in *C. gibelio* and around 0.48 (approximately 1/2) in *C. auratus* (Figure 1C). These data indicate that *C. gibelio* usually has three alleles, and *C. auratus* has two alleles for each *klhl* gene, consistent with their polyploidy (Figure 1A). In addition, *klhl1* was likely lost in the ancestor of teleost, and *klhl43* was likely a new member that emerged in ray-finned fishes after diverging from tetrapods.

### 2.2. Evolution of klhl Gene Family in Carassius

To clarify orthologous relationships, we constructed an ML phylogenetic tree containing all Klhl members identified from zebrafish, *C. auratus*, and *C. gibelio* assemblies. Of a total of 55 gene branches (Figure 2A), 28 *klhl* genes showed a ratio of 1:2:2 among zebrafish, *C. auratus*, and *C. gibelio* (e.g., *klhl33* in group Ⅲ, *klhl6* in group Ⅳ, and *klhl4* in group V). Gene expansions in a special fish were also observed, including 10 genes in zebrafish (e.g., *klhl2* in group V and *klhl9* in group III), one in *C. auratus* (*klhl30* in group Ⅳ) and two in *C. gibelio* (*klhl8* and *klhl39b* in group V). The rest were concerned with gene loss. For example, *klhl27* (group V) and *klhl34* (group III) were missing in zebrafish, and 15 genes had only one duplicate in both *Carassius* species (e.g., *klhl3*, *klhl7*, and *keap1b* in group V). However, we did not find any special gene loss only in *C. auratus* or in *C. gibelio*.

To clearly present evolutionary trajectories of the *klhl* gene family in *Carassius*, we explored their chromosome localization (Appendix A) and classified them into four categories (Appendix A). Consistently, 35 out of 55 *klhl* genes (category I) had two duplicates that just resided at a pair of homeologous chromosomes, both in *C. auratus* and *C. gibelio*, forming a 2:2:1 relationship with the zebrafish homolog (Figure 2B, red lines). Therefore, these two *klhl* genes were a pair of homeologs named by adding ‘-A’/’-B’ after the symbol of their zebrafish orthologue (e.g., *klhl1a-A* and *klhl1a-B*). Besides possessing two homeologous duplicates, *klhl30* and *klhl39b* (category II) appeared to have more gene copies (Figure 2B, blue lines). These extra gene expansions only occurred in one species and may come from small-scale duplication. The third category contained 14 *klhl* genes that displayed biased retention/loss of 1 homeolog (Figure 2B, green lines). Except for *keap1b*, the biased retention/loss of other genes was shared by the two *Carassius* species and not asymmetrical to subgenome (five A-retained genes and six B-retained genes). The remaining four *klhl* genes with particularly uncertain cases were grouped in category IV (Figure 2B, gray lines).

We further investigated the association of the biased retention/loss of homeologs with the paralogs that were duplicates that generally resulted from the Ts3R and were usually named with the symbol of their mammalian orthologue followed by ‘a’/’b’ (e.g., *klhl1a* and *klhl1b*). A total of 42 *klhl* members with reliable homeologous and paralogous relationships were classified into four types (Table 1): Six *klhl* members showed no biased retention/loss of either paralog and homeolog (Type I); Four *klhl* members that did not lose any paralog had lost at least one homeolog (Type II); and Type III contained the largest number (22) of *klhl* members, of which one paralog had lost, but the remaining paralogs maintained all homeologs. The last type included 10 members, of which both paralogs and homeologs occurred biased retention/loss. Therefore, the paralogs that experienced biased retention/loss after the Ts3R tended to maintain homeologs after the allotetraploidy in *Carassius*.

### 2.3. Expression Atlas of klhl Genes in Carassius

To preliminarily infer their functions, we first investigated *C. auratus klhl* (*Caklhl*) gene expression across eight adult tissues and seven embryogenesis stages. Transcriptome results were verified as the consistent gene expression results of almost all 12 randomly selected *klhl* genes that were obtained by quantitative reverse transcription PCR (qPCR) analysis (Appendix A). Hierarchical clustering analysis showed that their expression patterns could be categorized into two distinct clusters (I and II) in adult tissues (Figure 3A): Cluster I included 47 genes that were expressed ubiquitously and averagely in most of the analyzed tissues; and Cluster II, containing the rest of the 49 genes, was subdivided into three subclusters (II1, II2, and II3) and mainly showed a tissue-preference expression pattern or low expression level. Genes in II1, such as *klhl31-A*, *klhl31-B*, *klhl41b-A*, and *klhl41b-B*, had much more expression in the muscle and/or heart than in other tissues. Genes in II2 showed a moderate expression level in one or several of the analyzed tissues. For example, *klhl14-A* and *klhl14-B* were specifically expressed in the brain, *klhl32-B* and *klhl5-A* were expressed in six tissues except the liver and muscle, and *klhl26-B* was mainly expressed in the ovary. The last subcluster contained eight genes that exhibited low expression level in almost all analyzed tissues. Similarly, based on the dynamic expression abundance during embryogenesis, *Caklhl* genes were also divided into two clusters (I and II) and each cluster was grouped into three subclusters. All 55 genes in cluster I showed abundant transcripts throughout embryogenesis, and most of them (e.g., *klhl18s*, *klhl4s*, and *klhl13s*) displayed the highest transcript abundance at the early stages (Figure 3C). Compared with cluster I, genes in cluster II were expressed at a lower level. Subcluster II1 genes gradually increased transcription abundance during embryogenesis. Consistent with the high expression level in the adult heart, a large amount of transcripts from both homeologs of *klhl31*, *klhl40a*, *klhl41b*, and *klhl43* were detected from Prime-5 stage with the heart beating. Subcluster II2 included eight genes with a lower transcription level, and the transcripts of subcluster II3 genes were almost not detected (Figure 3C).

*C. gibelio* and *C. auratus klhl* orthologs showed very high sequence similarity: 68 out of 99 ortholog pairs appeared to have higher identities than 98% in the amino acid sequence, of which 20 ortholog pairs are exactly the same (Appendix A). Also, consistent with their close evolutionary relationship, most of the *Cgklhl* genes had a very similar expression pattern as *Caklhl* orthologous genes (Figure 3). But there were still several genes displayed with an obvious difference between them. Compared to *C. auratus* adult tissues, for example, the *Cgklhl11-A* expression level decreased in all tissues, the *Cgklhl16-B* and *Cgklhl37a-B* expression levels were upregulated in the kidney, the *Cgklhl42-A* and *Cgklhl38a-B* expression levels increased in multiple tissues, and the expression of *Cgklhl23.1-A* was upregulated in the kidney, hypothalamus and pituitary (Figure 3B). Meanwhile, *klhl42-A*, *klhl29a-A*, and *klhl11-A* also showed differential expression patterns during embryogenesis (Figure 3D). Interestingly, except *Cgklhl38a-B*, all these differentially expressed genes showed very high identity, ranging from 98.8% to 100% (Appendix A). Compared with the homologs in *C. auratus*, all *C. giblio*-specific duplicated copies (*Cgklhl39-B1/B2* and *Cgklhl8-1/2*) showed similar expression patterns, and the total number of transcriptions were also comparable, suggesting their function may be conserved.

### 2.4. Varied Expression Bias of klhl Homeolog Pairs in Carassius

In polyploid, duplicated genes generally altered their expression patterns [17]. For combined analysis, in adult tissues, 68% of *Caklhl* and 47% of *Cgklhl* homeolog pairs showed balanced pattern (Figure 4A), with similar relative transcript abundance between homeologs A and B. The remaining *klhl* homeolog pairs belonged to homeolog-dominant patterns, with higher or lower expression from one homeolog relative to another. Nine *klhl* homeolog pairs showed consistent homeolog-dominant patterns (four with A-dominant and five with B-dominant) in *Carassius* (Appendix A). There were 11 homeolog pairs that expressed were bias expressed (six with A-dominant and five with B-dominant) exclusively in *C. gibelio*, whereas only three pairs (all with B-dominant) were *C. auratus*-specific (Figure 4B). However, during embryogenesis, a few more *C. auratus klhl* homeolog pairs showed bias expression in *C. auratus* than in *C. gibelio* (Figure 4A). Besides seven shared pairs (Appendix A), there were eight and six homeolog-dominant pairs exclusively in *C. auratus* and *C. gibelio*, respectively (Figure 4B). Among them, an interesting gene is *klhl29a*, which displayed homeolog B-dominant expression pattern in *C. auratus* but an A-dominant expression pattern in *C. gibelio*. In addition, these homeolog-dominant pairs usually expressed bias in multiple tissues or developmental stages rather than only one (Figure 4B and Appendix A).

We further investigated the expression bias of *klhl* genes in an individual sample. Consistent with the combined analysis, *C. auratus* and *C. gibelio klhl* genes both showed obvious homeolog B-dominant expression patterns in all adult tissues (Figure 4C, indicated by the medians). Except in the muscle and brain, *Cgklhl* genes displayed more homeolog B-dominant expression patterns in more tissues than *Caklhl* genes. In contrast, the expression levels of *klhl* homeologs were more balanced in *C. gibelio* during early embryogenesis (Figure 4D, E1–E4). From the 8-somite stage (E5), *klhl* genes displayed consistently homeolog B-dominant expression patterns in the two species. This may be related to the organogenesis and/or myogenesis in the late stages, during which the gene expression repertoire might be similar as in adult tissues. In addition, five homeolog pairs displayed a shift of expression dominance temporally during embryogenesis (Figure 4E). Except for *klhl17a*, four pairs showed an expression-bias shift both in *C. auratus* and *C. gibelio*, but their shifted patterns were not exactly the same. Four *Caklhl* genes shifted their expression bias between blastula (E2) to shield (E3) stages, while two genes (*Caklhl41a* and *Cgklhl28*) shifted between Prime-5 (E6) and Pec-fin (E7) stages.

### 2.5. klhl Expression in Response to Herpesvirus Infection in C. gibelio

Polyploids usually show enhanced tolerance to environmental or other stress [36]. To explore the functional divergence of *klhl* genes after autotriploidy in *C. gibelio*, we examined their expression in three *C. gibelio* gynogenetic clones infected by *Carassius* herpesvirus 2 (*Ca*HV-2), one of the main pathogens in aquaculture of crucian carp [37,38]. The three clones diverged from a *C. gibelio* ancestor about 0.86 million years ago [19] but showed distinct herpesvirus resistance (clone A^+^: susceptible to *Ca*HV-2; clone F: weakly resistant to *Ca*HV-2; clone H: highly resistant to *Ca*HV-2) [39]. Most of the *klhl* genes (70 out of 98) showed similar expression patterns among the three clones (Figure 5A). However, differentially expressed *klhl* genes (DEGs) in response to *Ca*HV-2 infection showed an interesting pattern change among them. Clone H had 13 clone-specific DEGs, whereas clones A^+^ and F had only two clone-specific DEGs. Except for five downregulated DEGs (*klhl4-A*, *klhl6-A*, *klhl9-B*, *klhl24a-B*, and *klhl39a-B*) shared by all three clones, clone F shared eight and two DEGs with the same expression change with clone A^+^ and H, respectively, whereas clone A^+^ and H shared zero DEGs with similar expression change. In contrast, clone A^+^ and H had four DEGs that showed opposite expression changes after *Ca*HV-2 infection (e.g., *klhl5-A*, *klhl17b-B1*, *klhl24a-A*, and *klhl39b-B1*). Correlation analysis indicated that clone A^+^ and H had the most distinct expression pattern for *klhl* genes in response to herpesvirus infection (*r* = −0.05), while clone F is in the middle of them (r = 0.77 for clone A^+^ and r = 0.13 for clone H) (Figure 5B).

Many differentially expressed *klhl* genes have been reported to be associated with cellular inflammation and immune regulation against viral infection [2,4]. For example, *keap1a*, a key sensor of oxidative stress and virus infection [40], was specifically upregulated in infected fish of clone H (Figure 5C), possibly related to the high resistance against *Ca*HV-2; *klhl12B*, a suppressor to enterovirus-mediated translation [41], was downregulated in clone A^+^ and F (Figure 5D). Considering that Klhl proteins usually participate in the ubiquitination of substrate proteins via interacting with Cullin3 [4], other differentially expressed *klhl* genes may also be associated with the host’s pathological response to viral infection. Therefore, we speculate that the dynamic patterns of the differential expression of *klhl* genes would contribute to the different antivirus response among *C. gibelio* clones.

### 2.6. klhl Alleles in C. gibelio

To study variations among triploid homologs, we tried to identify *Cgklhl* alleles using a haplotype-resolved assembly of *C. gibelio* (see Materials and Methods) that contained 150 chromosomes and a read coverage analysis based on the assembly with 50 chromosomes (see details in methods). Of the total 98 *Cgklhl* genes, 90 genes possessed three alleles, whereas seven genes had two alleles and their average read depths (around ~36) were approximately 2/3 of the average depth for other *Cgklhl* genes (Appendix A and Figure 6A). Unexpectedly, *Cgklhl18-B* had only one allele, although its read depth was equal to the depth of genes with three alleles (Figure 6A): sequence comparing to the assembly with 150 chromosomes indicated that this gene had three other nonchromosomal homologous fragments that did not possess genes (Appendix A). We verified the alleles of all biallelic and monoallelic genes as well as 10 randomly selected triallelic genes, by PCR and Sanger sequencing using tissue cDNA mixture and genomic DNA templates. Therefore, allele loss of *C. gibelio* would be mainly caused by large fragment deletion.

Allelic sequence identities of the *Cgklhl* genes ranged from 88.12% to 100.00% (Appendix A), and they were not significantly different between the genes belonging to subgenomes A and B (Appendix A). 81 out of 97 genes showed very high interallelic identities, higher than 99%. There were four genes (*klhl4-A*, *klhl4-B*, *klhl12-A*, and *klhl29b-A*) that showed interallelic identities lower than 90%, owing to small DNA fragment deletion in one of the alleles (Appendix A), but their structures of protein among alleles were not obviously different. Instead, the other three genes with high interallelic nucleotide identities had one allele that showed different protein domain structures (Figure 6B). One allele of *klhl34-A* lost most domains due to the small indels that caused a frame shift of the allele, whereas the single nucleotide polymorphisms resulted in missing one Kelch domain in one *klhl42-A* allele and two Kelch domains in two *klhl23.2-A* alleles.

### 2.7. Bias Allelic Expression of Cgklhl Genes

In general, the two alleles of diploid are expressed at similar levels, but some of them display unequal expression, referred to as allelic-specific expression (ASE) [42,43]. To understand how genetic interactions among alleles of amphitriploid influence gene expression, we analyzed the ASE of *klhl* genes in *C. gibelio* using STAR pipeline. The relative expression of three alleles determined a triad position in a ternary plot (Figure 7A, top left), which allows the analysis of allele expression bias. Accordingly, we defined three expression bias categories: a balanced category with similar transcript abundance among the three alleles (all alleles > 0.25); and suppressed or dominant categories, where one allele shows lower (one allele < 0.25, two alleles > 0.25) or higher (one allele > 0.5, two alleles < 0.25) transcript abundance than those from another two alleles. We first performed a global analysis that combined data across all seven adult tissues or eight developmental stages. In adult tissues, 10, 46, and 24 *Cgklhl* genes were classified into the balanced, suppressed, and dominant categories, respectively (Figure 7A); however, during embryogenesis, only two genes belonged to the balanced category and the dominant category included 45 genes (Figure 7A). The separated analysis for the individual sample also confirmed the allele expression bias, in which no gene belonged to the balanced category during the first five developmental stages (Figure 7B). A high proportion of unbalanced allele expression indicates that the alleles of most *klhl* genes have conspicuously diverged, especially during embryogenesis.

To explore the plasticity of the allele triads, we investigated their expression variation across the 15 transcriptome samples by calculating the mean distance from each triad position to the global average center (Figure 7C, left). In adult tissues, the mean distances of the 3 categories were around 0.097, 0.085, and 0.0081, respectively, and were not significantly different among them (all *p*-values > 0.1, two-sample *t*-test) (Figure 7C, right). Interestingly, the mean distances of all categories during embryogenesis (the medians were 0.31, 0.19, and 0.11, respectively) were significantly larger than those in adult tissues (all *p*-values < 0.05, two-sample *t*-test) (Figure 7C, right). These results indicate that the relative expression patterns of the three alleles are rather stable among adult tissues but dynamic during embryogenesis. Then, we hypothesized that the dynamic during embryogenesis might be associated with the allele dominance shift (ADS). To test this, we defined an allele as dominant when it accounts for more than 50% of the total transcript abundance in any of the samples, and examined the expression patterns of all dominant alleles. As shown in Figure 7D, no second allele was found to be dominant for all *Cgklhl* genes in all sampled adult tissues, whereas 29 out of 84 filtered triallelic genes had two dominant alleles for different developmental stages. Therefore, the ADS is indeed common during embryogenesis, suggesting that many alleles might function in a sequential manner. In addition, the ADS mainly took place between the 4-cell and blastula stages, corresponding to the maternal to zygotic transition, and between the 8-somite and 1-dpf stages, corresponding to the organogenesis.

## 3. Discussion

Whole genome sequencing and comparative genomic analyses have proved the impacts of WGD on genomic diversity, variability, and complexity [13,15,44,45]. The direct result of WGD is producing huge duplicates at the genome level. How the duplicates evolve their structure and expression for adapting to coexistence and for harmonizing is fascinating and needs extensive studies. It is critical for a new polyploid to alleviate conflicts of duplicates from genome mergers. Biased genomic changes may help mitigate the initial chaos and are essential for the subsequent rediploidization process [18,45]. The most likely evolutionary fate of redundant duplicated genes is loss of one duplicate, more than 70.0–80.0% of duplicated genes after Ts3R were lost in the first 60 million years [46]. Due to their relatively short evolutionary history, common carp (*Cyprinus carpio*) and goldfish did not experience extensive homeologous gene loss, but asymmetric subgenome evolution has been mentioned in both allotetraploids [47,48]. In this study, we found that 75% of *klhl* members have lost one paralog after Ts3R and 14 out of 55 *klhl* genes have lost one homeolog after 4R (Appendix A and Figure 1B). The *klhl* paralogs that experienced biased retention/loss are more likely to maintain homeologs. Dosage balance is likely a major driver for the patterns of duplicated gene retention and loss after polyploidization.

WGD provides flexibility to facilitate adaptive traits of species via tissue-specific expression or sub- or neofunctionalization of massively duplicated genes [15,49]. Differential homeolog expression, defined as homeolog bias, is a ubiquitous feature of plant polyploids [15,49]. For example, about 30.0% of hexaploid bread wheat (*Triticum aestivum*) homeolog triads (AABBDD subgenome) showed homeolog-dominant or homeolog-suppressed expression patterns [15,46]. However, besides expression bias, variant subgenome expression divergence processes have been observed in common carp and goldfish [50,51]. Versatile expression divergence is also found in the *Carassius klhl* gene family. Although expression dominance to homeolog B is in evidence, balanced homeolog pairs remain in the majority (Figure 4). Compared to adult tissues and late developmental stages, *klhl* homeolog pairs show more divergence in early developmental embryos, and some homeolog pairs even shift their expression bias during embryogenesis (Figure 4E). Interestingly, some homeolog pairs show different expression bias between *C. auratus* and *C. gibelio* although they share the same allotetraploidy event. These findings may be the result of the homeologous recombination and/or from the genome shock of the recent autotriploidy in *C. gibelio*. Polyploidy has been considered as an evolutionary and ecological force in stressful times or harsh environments, and polyploids often show enhanced tolerance to abiotic or biotic stress, including attacks by pathogens [36]. Relative to their ancestors, the increased tolerance of polyploids is thought to be attributable to several factors, such as genome buffering, gene expression altering, epigenetic remodeling, gene regulatory network rewiring, and sub- and/or neofunctionalization of duplicate genes [18,52]. The extra autotriploidy may endow *C. gibelio* with abilities for additional adaptive plasticity. Although most of the *klhl* genes had analogous expression patterns between *C. gibelio* and *C. auratus* (Figure 3), even including species-specific duplicated copies (*Cgklhl39-B1/B2* and *Cgklhl8-1/2*) that also showed conserved expression patterns, there are still some genes that have undergone significant changes (Figure 3). Furthermore, this adaptive plasticity was also demonstrated by the divergent expression pattern among different *C. gibelio* gynogenetic clones infected with *Ca*HV-2 (Figure 5). Although the three clones diverged from a *C. gibelio* ancestor about 0.86 million years ago [19], *klhls* expression changes in response to viral infection showed a clear correlation with different viral resistance among the three clones. Given the function of *klhl* gene in protein ubiquitination and the pathogenesis of diseases [4], the dynamic and variable expression divergences that improve functional plasticity and flexibility of duplicates may play important roles in evolutionary innovation for ecological adaptation, which awaits further investigation.

*C. gibelio klhl* genes are very similar in sequence and expression as *C. auratus* (Figure 3 and Appendix A). In addition, no significant within-individual allelic diversity or decay were found in *C. gibelio* (Figure 6), although prolonged unisexual gynogenesis should naturally cause high homologous differentiation and accumulation of mutations [53]. The gene introgression/exchange with sympatric sexual species, especially with *C. auratus*, would help to shape the genome of *C. gibelio*. Recently, we found that changes in ploidy drive unisexual and sexual reproduction transition, and thereby facilitate genetic exchange from *C. auratus* to *C. gibelio* lineages [54]. This phenomenon has also been observed in many unisexual organisms, such as salamanders [54], Iberian-Roach [55], and water frogs [56] that can benefit from sampling the gene pools of the sexual relatives [57]. Moreover, homologous recombination during oogenesis may grant *C. gibelio* abilities to clean mutation and to keep genome stable [19]. Therefore, to achieve evolutionary longevity, *C. gibelio* might employ multiple strategies (from the interspecies and intraspecies) that not only refresh its genome but also increase the population genetic diversity.

Although homologous divergence is limited, the expression among alleles diverges somewhat from exclusive expression of one or two alleles to subtle expression differences among alleles (Figure 7). Interactions among alleles range from buffering effects when alleles are functionally redundant to dominance effects when variations in single/two alleles can lead to dominant phenotypes. This allelic expression imbalance is usually due to cis-acting genome variations because environmental or trans-acting elements commonly affect all alleles equally [42]. Besides transcription factor binding, chromatin accessibility mediated by epigenetic imprinting [58], random monoallelic expression, posttranscriptional regulation including nonsense-mediated decay, and alternative splicing [59,60] may also lead to allele-specific imbalance. Recently, another haplotype-resolved assembly of *C. gibelio* from the Olsa river close to Ostrava was reported [59,60]. Therefore, it will be valuable to study how haploids coordinate to evolve in this attractive fish and to compare these evolution features at the differential lineage background in the future.

In conclusion, we focused on the lineage specific expansion of the *klhl* gene family, which exerts a variety of critical functions by regulating ubiquitination of specific substrates, and elaborated the paralog/homeolog/homolog diversification, biased retention/loss, and differential expression across adult tissues and embryogenesis in *Carassius*. The current study will contribute to our understanding of fish *klhl* genes and provides a paradigm showing the evolutionary process of duplicated genes in complex genomes that experienced multiple rounds of polyploidization events. This understanding should increasingly inform ecological analyses of adaptation and permit an enhanced appreciation for genomic evolution of polyploidy-fueled diversification.

## 4. Materials and Methods

### 4.1. Identification and Characterization of klhl Family Genes

The genome assembly of *C. auratus* (GCA_019720715.2) and *C. gibelio* (GCA_019843895.2), as well as of other 11 species, including Cephalochordata amphioxus (*Branchiostoma belcheri*) (GCA_001625305.1), Sea lamprey (*Petromyzon marinus*) (GCA_010993605.1), elephant shark (*Callorhinchus milii*) (GCA_000165045.2), lobe-finned fish African coelacanth (*Latimeria chalumnae*) (GCA_000225785.1), chicken (*Gallus gallu*) (GCA_000002315.5), human (*Homo sapiens*) (GCA_000001405.28), Spotted gar (*Lepisosteus oculatus*) (GCA_000002035.4), zebrafish (*Danio rerio*) (GCA_000002035.4), fugu (*Fugu rubripcm*)(GCA_901000725.2), rainbow trout (*Oncorhynchus mykiss*) (GCA_901000725.2), and Atlantic salmon (*Salmo salar*) (GCA_905237065.2) were downloaded from available databases including Ensembl (http://asia.ensembl.org/index.html, accessed on 4 August 2021) and NCBI (https://www.ncbi.nlm.nih.gov/, accessed on 4 August 2021). Haplotype-resolved assembly of gibel carp was deposited in NGDC (https://ngdc.cncb.ac.cn/) under the accession number of PRJCA011199. First, the *klhl* members were identified through the annotation file for each genome. Second, Klhl protein sequences of humans and zebrafish were used for homology-based gene prediction using TblastN (blast-2.2.26), and the predicted genes were merged to the initial *klhl* genesets. Third, the hidden Markov model profiles of Klhl BTB (PF00651) and BACK (PF07707) domain (http://pfam.xfam.org/, accessed on 30 August 2021) were utilized as a query to verify and supplement the *klhl* genesets by using the HMMER 3.0 program with default parameters. The putative *Cgklhl* genes were named based on their reciprocal best homologs of humans and/or zebrafish. The complete cDNA sequences of 96 *Caklhl* genes (GenBank accession no. OL598607-OL598707) and 98 *Cgklhl* genes (GenBank accession no. OL677882-OL677984) were deposited in GenBank. The length of CDS/protein, isoelectric pointprotein (pI), and molecular weight (MW) of *klhl* genes were calculated by biopython module of python version 3.9.0.

### 4.2. Phylogenetic Analysis

Multiple alignments of the full length of Klhl amino acid sequences of *C. auratus*, *C. gibelio*, and *D. rerio* were achieved using ClustalW program of MEGA software (version 7.0) [61] with Neighbour-Joining (NJ) method and 1000 bootstraps. Subsequently, the phylogenetic tree was constructed and visualized using Figtree v1.4.3 (http://tree.bio.ed.ac.uk/software/Figtree/, accessed on 27 April 2021).

For allelic phylogenetic analysis, 90 triallelic *klhl* genes were selected to align with the orthologous *Caklhl* and *Drklhl* genes and construct phylogenetic trees with Neighbour-Joining (NJ) method and 1000 bootstraps.

### 4.3. Synteny Analysis

MCScanX was used to identify syntenic blocks between *C. gibelio* genome (GCA_019843895.2), *C. auratus* genome (GCA_019720715.2), and *D. rerio* genome (GCA_000002035.4) with the parameters of -a -e 1e-5 -u 1 -s 5. First, we used BLASTP to align proteins among the three genesets with the parameters “1 × 10^−5^”. The alignments were then subjected to MCScanX to determine syntenic blocks and visualized via TBtools (version 1.082) [62].

### 4.4. RNA-seq Samples and Gene Expression Analysis

RNA-seq data of eight adult tissues, embryos at seven developmental stages, and healthy/*Ca*HV-2-infected head kidneys of three clones were downloaded from NCBI (PRJNA834570, PRJNA836313, PRJNA833750, PRJNA837728, and SRP096800). RNA-seq reads were mapped to the references using STAR [63] and the expression level was calculated using reads per kilobase per million mapped reads (RPKM). Pearson correlation coefficients were calculated in R 3.3.2.

The homeolog A/B expression ratio was calculated according to (log2 ((RPKM.A + 0.01)/(RPKM.B + 0.01))). The homeologous pairs were removed when the RPKM value of any gene was <5. The ratio of A to B expression more than 2 or less than 0.5 was defined as homeolog A- or B-dominant. For expression-bias shift analysis, A-dominant or B-dominant was defined when the ratio of A/B expression (Log2) was >0.1 or <−0.1.

Quantitative reverse transcription PCR (qPCR) was performed as previously described [64]. Heart and brain were sampled from adult *C. gibelio*. *Ca*HV-2 infection and sample collection were performed as previously described [65] Specific primes for each gene were listed in Appendix A. *Actin* was selected as the optimal reference gene. The relative gene expression levels were calculated with 2^−ΔΔCT^ method [66]. All the samples were analyzed in triplicate.

### 4.5. Polyploidy Evaluation and Allele Identification of Cgklhl Genes

BWA (Version 0.7.12-r1039) was used to map the Illumina reads of *C. auratus* and *C. gibelio* that we submitted to NCBI previously (accession numbers: OP253321-OP253605) to their genomes respectively and sorted to obtain the bam files by SAMtools (Version 1.4). The read depths for *klhl* genes were calculated using BamDeal (Version 0.25).

The CDS of *Cgklhl* genes were aligned to the haplotype-resolved assemblies (wang submitted) using blastn of TBtools (version 1.082). Combined with the ratio of average read depths of each gene to the average depth for other *Cgklhl* genes, to predict the number of alleles of *Cgklhl* genes, 1 corresponds to 3 alleles, 2/3 corresponds to 2 alleles, and 1/3 corresponds to 1 allele. We identified 285 *klhl* alleles for *C. gibelio*. Finally, PCR and 30 clones were sequenced to validate the alleles of randomly selected genes.

### 4.6. Allele-Specific Expression Analysis

Ninety triallelic *klhl* genes were selected for allele specific expression analysis. We defined a triad as expressed when at least 1 allele had the expression level >1 RPKM. After filtering, 80 and 84 triallelic *klhl* genes were left in the adult tissues and in the embryos, respectively, for further analyses. The relative expression of each allele across the triad was normalized by the total expression level of the gene. The triad was defined as ‘Dominant’ when the relative expression of one allele was >0.75, as ‘Suppressed’ when the relative expression of one allele was <0.25 and any of the other two alleles were <0.75, and as ‘Balanced’ when the relative expressions of all alleles were >0.25. In order to analyze the allele express shift, the allele expression values of the three alleles for all tissue or embryo samples were normalized firstly, and the allele with the highest expression level in a sample was defined as the dominant allele in this sample. By the end, the normalized expression levels of all dominant alleles were selected to drawn for all tissue or embryo samples. We calculated the Euclidean distance using the rdist function from R 3.3.2.

### 4.7. The Domains of Klhl Alleles

In order to predict protein conserved domains, the *Cg*Klhl protein sequences were submitted to Pfam [67], NCBI-CDD [68] and SMART [69], and visualized by TBtools (version 1.082). 

## 5. Conclusions

We explored the structure and expression evolutionary trajectories of *klhl* gene family in polyploid *Carassius*. Paralog/homeolog/homolog diversification, biased retain/loss, and versatile differential expression were elaborated in *Carassius klhl* gene family. The current study will contribute to our knowledge on fish *klhl* genes and provides a paradigm showing evolutionary process of duplicated genes in complex genomes that experienced multiple rounds of polyploidization events. This understanding should increasingly inform ecological analyses of adaptation and permit an enhanced appreciation for genomic evolution of polyploidy-fueled diversification.

## Figures and Tables

**Figure 1 ijms-24-08367-f001:**
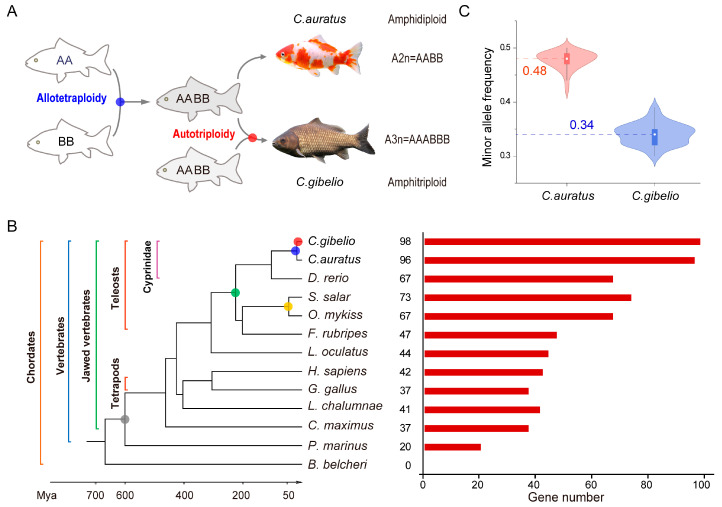
*klhl* gene families and polyploidy events in *Carassius* complex and other 11 animals. (**A**) Two polyploidy events occurred during *Carassius* evolution. (**B**) Number of *klhl* genes identified from the amphihaploid genomes of *Carassius* complex and other 11 animals. Grey, green, yellow, blue, and red cycles indicate the 1R/2R, Ts3R, salmon autotetraploidy, Cyprininae allotetraploidy, and *Carassius* autotriploidy events, respectively. (**C**) Violin plot for minor allele frequency of SNP sites in *Carassius klhl* genes. The line in the middle of each boxplot represents the median of the dataset.

**Figure 2 ijms-24-08367-f002:**
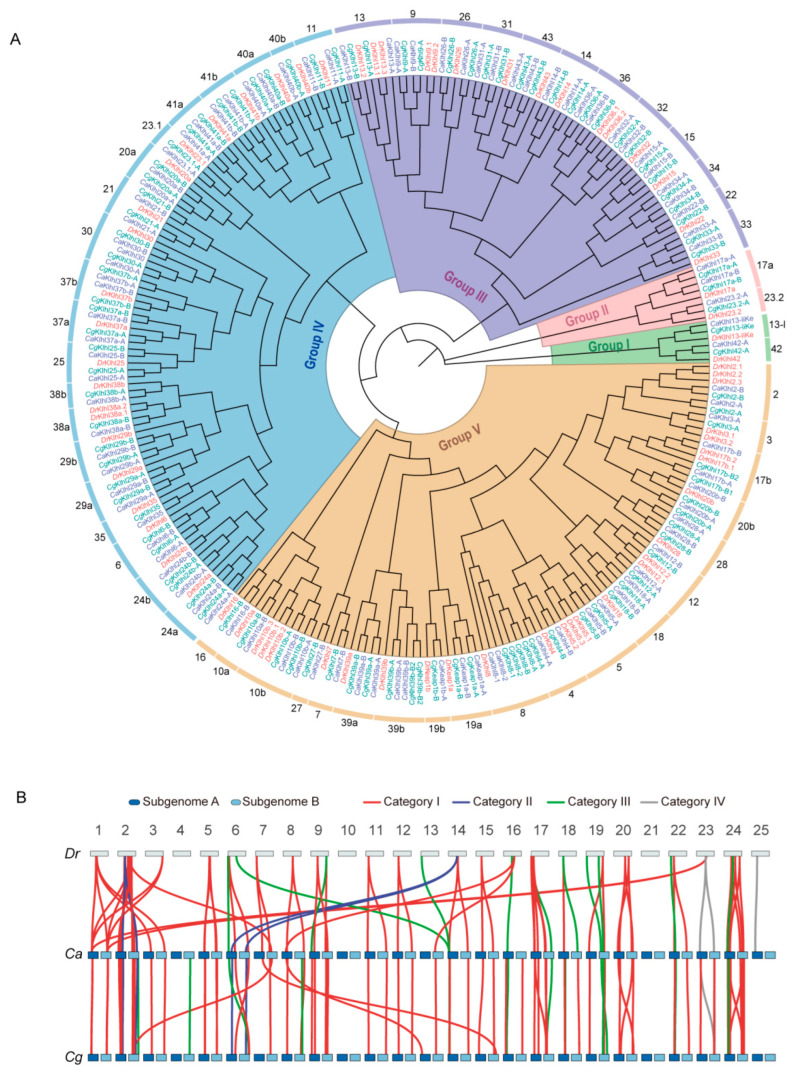
Phylogenetic and syntenic analyses of *klhl* gene families. (**A**) ML phylogenetic tree containing all *Klhl* members of *D. rerio* (*Dr*), *C. auratus* (*Ca*), and *C. gibelio* (*Cg*). (**B**) Synteny of *klhl* genes among *D. rerio* (*Dr*), *C. auratus* (*Ca*), and *C. gibelio* (*Cg*). Orthologous genes were classed into four categories and linked by different color lines.

**Figure 3 ijms-24-08367-f003:**
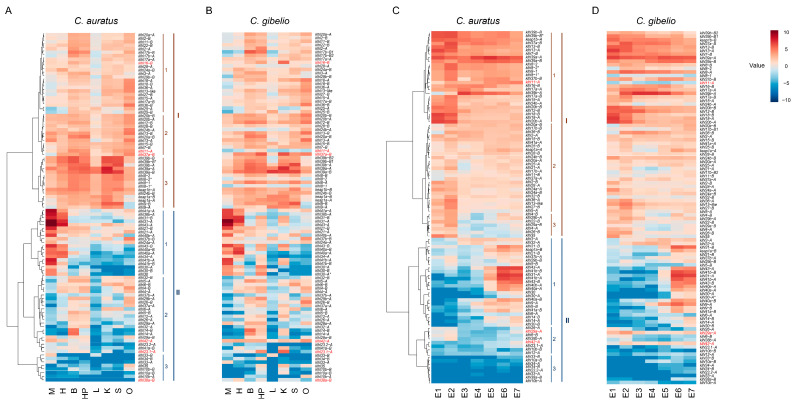
Heatmaps depicting *Carassius klhl* gene expression across eight adult tissues (**A**,**B**) and seven embryogenesis stages (**C**,**D**). The expression abundances were normalized by taking log2 of RPKM values, and the main hierarchical clusters were lined out on the left. M, Muscle; H, Heart; B, Brain; HP; Hypothalamus-pituitary; L, Liver; K, Kidney; S, Spleen; O, Ovary; E1, 4-cell stage; E2, Blastula stage; E3, Shield stage; E4, Bud stage; E5, 8-somite stage; E6, Prime-5 stage; E7, Pec-fin stage. Genes showing different expression patterns between *C. auratus* and *C. gibelio* are marked with red. For alignment, the missing genes are replaced by the homologous copies and indicated by asterisk.

**Figure 4 ijms-24-08367-f004:**
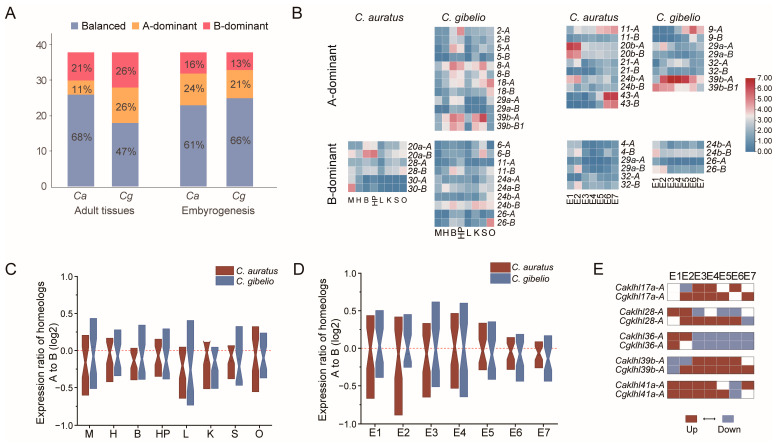
Expression divergence of *klhl* homeolog pairs in *Carassius* complex. (**A**) Combined analysis of gene expression showing balanced, homeolog A-dominant and homeolog B-dominant expression patterns for *C. auratus* (*Ca*) and *C. gibelio* (*Cg*). (**B**) Heatmaps of species-specific homeolog-dominant genes in the combined analyses for adult tissues and embryogenesis stages. (**C**,**D**) Expression ratio of homeolog A/B genes in the adult tissues (**C**) and the embryos at different developmental stages (**D**) of *Carassius* complex, zoomed in to show medians. M, Muscle; H, Heart; B, Brain; HP; Hypothalamus-pituitary; L, Liver; K, Kidney; S, Spleen; O, Ovary; E1, 4-cell stage; E2, Blastula stage; E3, Shield stage; E4, Bud stage; E5, 8-somite stage; E6, Prime-5 stage; E7, Pec-fin stage. (**E**) Expression trend of homeolog pairs with an expression-biased shift during embryogenesis. Ratio of A/B expression (Log2) is used for heatmap. Red tile (>0.1) represents A-dominant, while blue tile (<−0.1) represents B-dominant.

**Figure 5 ijms-24-08367-f005:**
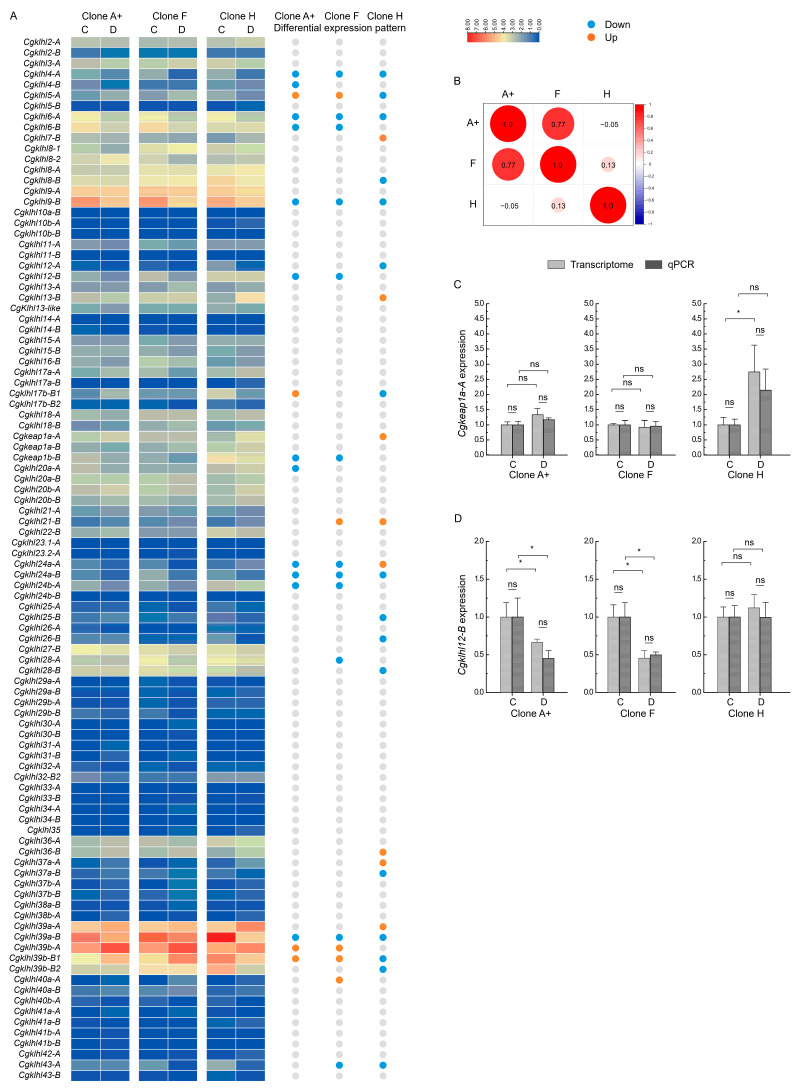
Expression of *klhl* genes in response to herpesvirus infection. (**A**) Heatmaps depicting expression of *klhl* genes in three *C. gibelio* clones before and after *Ca*HV-2 infection. Differentially expressed genes with upregulated and downregulated expression levels after *Ca*HV-2 infection are indicated by red and blue circles, respectively. C: Head–kidney tissues from healthy fishes. D: Head–kidney tissues from infected fishes. (**B**) Pearson correlation coefficients (r) for expression changes of DEGs among three clones. (**C**,**D**) Expression verification of *keap1a-A* (**C**) and *klhl12-B* (**D**) by qPCR. Data are presented by mean ± sd. ns.: no significance; * *p* < 0.05.

**Figure 6 ijms-24-08367-f006:**
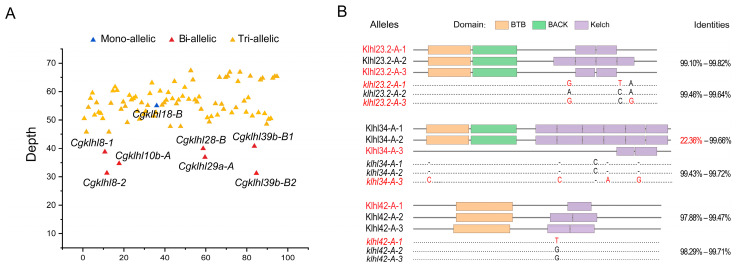
*klhl* alleles of *C. gibelio*. (**A**) Read depths of *Cgklhl* genes with different allele numbers. (**B**) Different nucleotide sequences and protein domain structures of the alleles. Nucleotide sequence and amino acid identities are shown on the right. Single nucleotide polymorphisms and indels affecting domain structure are presented and marked in red.

**Figure 7 ijms-24-08367-f007:**
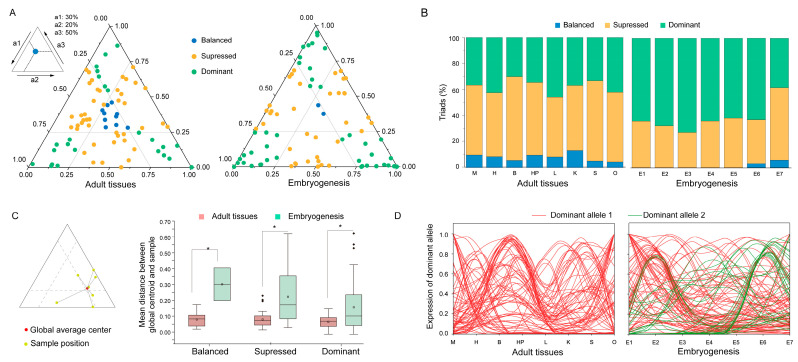
Allele-specific expression of *C. gibelio klhl* genes. Ternary plots showing relative expression of allelic triads in the adult tissues and the embryos at different developmental stages (**A**). Each dot represents an allelic triad with the coordinates determined by relative contribution of each allele to the total expression (an example is shown on the top left). (**B**) Proportion of triads in each category across the 15 samples. (**C**) Variation of triad expression patterns. Ternary plot shows an example for measuring the variation of triad expression (left). Box plots show the distance between the triad position of the individual samples and the global average position in each category in (**A**). The line in the middle of each boxplot represents the median of the dataset; the upper and lower edges of the boxplot indicate the third quartile and the first quartile, respectively; and the line extending from the edge is 1.5 times the interquartile range. Small dots indicate outliers. “*” indicates a *p*-value < 0.005. (**D**) Expression pattern of dominant allele for every sample across the 15 samples.

**Table 1 ijms-24-08367-t001:** Four types of *C. gibelio klhl* genes based on the biased retention/loss of homeologs of paralogs.

Type	Gene Name	Gene Number	Biased Retention/Loss of Paralogs	Biased Retention/Loss of Homeologs
I	*klhl20*, *klhl24*, *klhl29*, *klhl37*, *klhl39*, *klhl41*	6	No	No
II	*klhl10*, *keap1*, *klhl38*, *klhl40*	4	No	Yes
III	*klhl2*, *klhl4*, *klhl5*, *klhl6*, *klhl9*, *klhl11*, *klhl12*, *klhl13*, *klhl14*, *klhl15*, *klhl18*, *klhl21*, *klhl25*, *klhl26*, *klhl28*, *klhl30*, *klhl31*, *klhl32*, *klhl33*, *klhl34*, *klhl36*, *klhl43*	22	Yes	No
VI	*klhl3*, *klhl7*, *klhl13-like*, *klhl16*, *klhl22*, *klhl23.1*, *klhl23.2*, *klhl27*, *klhl35*, *klhl42*	10	Yes	Yes

## Data Availability

All relevant data are available from the corresponding authors upon request. There are no restrictions on data availability.

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
