# Peer review of "Genomic and Transcriptional Profiles of Kelch-like (klhl) Gene Family in Polyploid Carassius Complex"

_ijms, 2023, doi:10.3390/ijms24098367_

Round 1

Reviewer 1 Report

Gene duplication and divergence is one of the most important scientific questions in evolutionary biology. This study focused on the klhl family members in the genome of polyploid Carassius complex, which undoubtedly increased our understanding on the evolution of fish klhls and provides a paradigm showing evolutionary process of duplicated genes in complex fish genomes. However, the significance of the manuscript can be further highlighted if the authors could detect the expression of klhls in Carassius under environmental or other stress.

Reviewer 2 Report

I congratulate to the authors on the extensive and interesting study you have conducted here.

Gene composition and transcriptional activity of Kelch-like (klhl) gene family have been examined in two Carasius species - C. gibelio and C. auratus. Since polyploidy occurred in these species (allotetraploidy in C. auratus and autotriploidy in C. gibelio), the authors wanted to find out how this particular gene family evolved considering gene structure and function in these two species.
At the beginning the authors have given us basic information about the Klhl gene family, how many gene members are known and what kind of function the proteins encoded by these genes regulate. We are also introduced with WGD in general and WGD in Carassius species of interest.
The results are well organised and presented. The authors identified 96 klhl genes in C. auratus and 98 klh genes in C. gibelio, and 90 genes with three alleles were also recorded in C. gibelio. In addition, the evolution of klhl genes in the two species have been studied, taking into account gene expansion, gene loss, philogenesis and chromosome localisation. Comparative expression was also studied in eight adult tissues and seven embryogenesis stages, with particular attention to the relative expression of the three alleles of the klhl gene in C. gibelio to understand the genetic interaction between them. The authors identified three categories of allelic triads that differ in expression and an expression pattern that is relatively stable in adult tissues but dynamic during embryogenesis.
Finally, the evolutionary significance of whole genome duplication in general and in light of the results obtained in this study was properly discussed.

I warmly recommend “Genomic and transcriptional profiles of Kelch-like (klhl) gene 2 family in polyploid Carassius complex  “ for publication in the International Journal of Molecular Sciences.

I suggest some minor changes:

Figure 4. - part B and C in this figure look like one unit. I suggest either naming them as one unit (B) or adding missing information to both (0.00-7.00 scale for B; and B-dominant, A-dominant for C)
Line 161 - replace blastrula with blastula
Line 200 - replace blastrula with blastula
Line 216 - you have gastrula (E3) stages, but in line 161 and 200 E3 is called shield stage... You should unify this
Figure 6. - part A and B in this figure look like one unit. I suggest you name them as one unit (A)

Reviewer 3 Report

It is very difficult to follow the results in the manuscript. The authors provided larger amounts of data derived from bioinformatic analyses without a detailed description. In addition, some conclusions were not experimentally validated. Overall, the manuscript collected bioinformatic information on klhl genes in several teleost fish but lacked a clear link. It may be more appropriate for a specialized journal.

Importantly, it is not possible to have an idea on how this study helps understand the role of genome duplication on evolutionary innovation, transcriptional evolution, and ecological adaptation. 

There are also other concerns that make the manuscript unsuitable for publication:

1. The authors often use “homeologs/homeologous” and “paralogds/paralogous” in the text, but it is unclear and ambiguous when the klhl genes are homeolohous or paralogous.

2. It is not clearly explained how klhl genes and alleles in C. auratus and C. gibelio are expressed in adult tissues and during embryogenesis.

3. The relationship between conserved non-coding elements and allelic-specific expression was also unclear.

Round 2

Reviewer 3 Report

The authors have made some efforts to revise their manuscript, but I do not think this revised version is improved with respect to the concerns raised in my previous review. Again, bioinformatic data were not clearly described and some conclusions were not experimentally validated. In figure 5, they used heatmaps to compare klh1 expression in different clones, but these bioinformatic analyses were not validated by experimental data. The discussion on the evolutionary significance of klh genes is not convincing. Many genes show dynamic and differential expression. It is unclear how specifically klh1 polyploidy and dynamic differential expression patterns provide insights into transcriptional evolution

Round 3

Reviewer 3 Report

1. RT-qPCR results in figure S3 should be integrated in figure 5. Statistical significance should be indicated.

2. Lines 245-247, only two genes were verified by RT-qPCR, thus it is not appropriate to state that qPCR analysis was also used to verify the expression results obtained by transcriptome analysis

3. I see only six pairs of PCR primers but the authors examined 12 genes in figure S1.
